# Plasma membrane phosphatidylinositol (4,5)-bisphosphate promotes Weibel–Palade body exocytosis

Tu Thi Ngoc Nguyen, Sophia N Koerdt, Volker Gerke

**Weibel–Palade bodies (WPB) are specialized secretory organelles of endothelial cells that control vascular hemostasis by regulated, Ca$^{2+}$-dependent exocytosis of the coagulation-promoting von-Willebrand factor. Some proteins of the WPB docking and fusion machinery have been identified but a role of membrane lipids in regulated WPB exocytosis has so far remained elusive. We show here that the plasma membrane phospholipid composition affects Ca$^{2+}$-dependent WPB exocytosis and von-Willebrand factor release. Phosphatidylinositol (4,5)-bisphosphate [PI(4,5)P$_2$] becomes enriched at WPB–plasma membrane contact sites at the time of fusion, most likely downstream of phospholipase D1-mediated production of phosphatidic acid (PA) that activates phosphatidylinositol 4-phosphate (PI4P) 5-kinase γ. Depletion of plasma membrane PI(4,5)P$_2$ or down-regulation of PI4P 5-kinase γ interferes with histamine-evoked and Ca$^{2+}$-dependent WPB exocytosis and a mutant PI4P 5-kinase γ incapable of binding PA affects WPB exocytosis in a dominant-negative manner. This indicates that a unique PI(4,5)P$_2$-rich environment in the plasma membrane governs WPB fusion possibly by providing interaction sites for WPB-associated docking factors.**

## Introduction

Vascular homeostasis is delicately balanced to permit unrestricted blood flow but also prevent excessive leakage of plasma and blood cells in case of injury. Among the many factors that control this homeostasis is the pro-coagulant glycoprotein von-Willebrand factor (VWF), which recruits platelets to sites of vessel injury and thereby promotes the formation of a platelet plug. The main source of the highly pro-coagulant VWF is endothelial cells which store VWF in unique secretory granules known as Weibel–Palade bodies (WPBs). WPBs are considered lysosome-related organelles that undergo a complex maturation process involving formation at the trans-Golgi network and interactions with the endosomal system

(McCormack et al, 2017; Mourik & Eikenboom, 2017). Mature WPBs are anchored in the actin cortex as elongated, cigar-shaped organelles, their unique form being dictated by the tightly packed VWF tubules (Michaux & Cutler, 2004; McCormack et al, 2017). Endothelial stimulation, which occurs in response to blood vessel injury and leads to an intraendothelial Ca$^{2+}$ and/or cAMP elevation, triggers the exocytosis of mature WPB and thus the acute release of highly multimeric VWF into the vasculature (Schillemans et al, 2019; Karampini et al, 2020).

The final steps of regulated WPB exocytosis are complex and involve a number of factors that have been identified in the past. They include docking proteins such as Munc13-2 and Munc13-4, several Rab GTPases such as Rab3, Rab15, Rab32, and Rab46, and members of the SNARE family mediating the actual membrane fusion including syntaxin-2, syntaxin-3, VAMP-3, and VAMP-8 (Matsushita et al, 2003; Pulido et al, 2011; Zografou et al, 2012; Biesemann et al, 2017; Chehab et al, 2017; Schillemans et al, 2018; Holthenrich et al, 2019; Karampini et al, 2019; Miteva et al, 2019). Moreover, proteins associated with the SNARE and/or Rab machineries, such as synaptotagmin 5, synaptotagmin-like protein-4a (Slp4a), and Munc18-1, have been identified as positive regulators of evoked WPB exocytosis (Bierings et al, 2012; van Breevoort et al, 2014; Lenzi et al, 2019). Thus, similarities exist to other regulated exocytotic fusion events, for example, synaptic vesicle exocytosis in neurons, dense core granule exocytosis in neuroendocrine cells, and insulin secretion in pancreatic β cells (Jahn & Fasshauer, 2012; Röder et al, 2016; Gasman & Vitale, 2017). However, WPB exocytosis is also characterized by unique features, among other things the asymmetric shape and large size of the secretory granules most likely requiring specific docking/fusion scenarios, the apparent absence of a predocked or primed state of vesicles and the complex regulation that involves either Ca$^{2+}$- or cAMP-dependent pathways with signaling intermediates not well characterized to date (McCormack et al, 2017; Mourik & Eikenboom, 2017).

Lipids, both in the organelle membrane and the plasma membrane (PM), play important roles in exocytotic docking and fusion steps. They can support the actual bilayer fusion, for example, by locally changing curvature and can also serve as recruitment and/or activation platforms for proteins participating in the exocytotic

Institute of Medical Biochemistry, Center for Molecular Biology of Inflammation, University of Münster, Münster, Germany

Correspondence: gerke@uni-muenster.de

process. However, the precise role of fusion-supporting lipids, their potential accumulation and/or generation at exocytotic fusion sites, and their turnover in the course of the reaction are still largely enigmatic. Of particular relevance in this respect are two negatively charged phospholipids, phosphatidic acid (PA), and phosphatidylinositol (4,5)-bisphosphate [$PI(4,5)P_2$], which have been shown in different systems to promote exocytotic membrane fusion (Di Paolo & De Camilli, 2006; Ammar et al, 2013; Martin, 2015; Raben & Barber, 2017). PA also appears to be involved in $Ca^{2+}$-dependent WPB exocytosis evoked by histamine stimulation or following treatment with the B subunit of Shiga toxins as inhibition or depletion of the PA-generating enzyme PLD1 interferes with acute VWF release (Disse et al, 2009; Huang et al, 2012). However, the mode of action of PA in initiating and/or supporting WPB exocytosis is not known. $PI(4,5)P_2$, on the other hand, has not been linked to WPB exocytosis so far, but its involvement in regulated exocytosis in neurons and neuroendocrine cells has been documented (Eberhard et al, 1990; Hay et al, 1995; Di Paolo et al, 2004). Here, it is thought to function by binding to and regulating proteins that participate in the docking and/or tethering of exocytotic vesicles at the PM and the actual fusion step. Some $PI(4,5)P_2$-binding proteins, for example, Munc13-4, Slp4-a, and annexin A2, act as positive regulators of evoked WPB exocytosis suggesting that $PI(4,5)P_2$ could also affect regulated exocytosis in endothelial cells.

To assess the mechanism that underlies the role of PA and possibly other PM phospholipids in WPB exocytosis, we used phospholipid [PA, PI(4)P, $PI(4,5)P_2$, $PI(3,4,5)P_3$ and PS] binding probes and analyzed their distribution in response to secretagogue stimulation of WPB exocytosis with high spatiotemporal resolution. This approach took advantage of the exceptionally large size and asymmetric shape of WPB that allowed a straight-forward visualization of individual fusion events. Our data show for the first time that PA and $PI(4,5)P_2$ transiently accumulate at sites of WPB exocytosis and that this accumulation precedes the actual fusion event. General and acute $PI(4,5)P_2$ depletion as well as knockdown of the major $PI(4,5)P_2$–generating enzyme expressed in endothelial cells, PI4P-5 kinase γ, significantly interfere with histamine-evoked WPB exocytosis and VWF secretion. PI4P-5 kinase γ most likely acts downstream of PM PA as a mutant enzyme defective in PA binding is not recruited to the PM and inhibits evoked WPB exocytosis.

# Results

### $PI(4,5)P_2$ and PA but not PS accumulate at WPB–PM fusion sites

Phosphatidylinositol 4,5-bisphosphate [$PI(4,5)P_2$] has been implicated in exocytotic membrane docking and fusion processes and can possibly function synergistically with PA because the $PI(4,5)P_2$–synthesizing enzymes, PI4P 5-kinases (PIP5Ks), are stimulated by PA (Roach et al, 2012). Phosphatidylserine (PS) is another PM phospholipid that has been associated with vesicle-PM fusion events (Lou et al, 2017). Therefore, we recorded the behavior of these three phospholipids in the course of histamine-evoked WPB exocytosis by using multicolor total internal reflection fluorescence (TIRF) microscopy as well as live cell confocal microscopy of primary human endothelial cells (HUVECs) expressing different phospholipid-

binding domains as sensors. Specifically, PH-PLCδ1-YFP, the pleckstrin homology domain of PLCδ1 fused to YFP, was used as $PI(4,5)P_2$ sensor, Spo20p3-GFP, the membrane binding domain of the yeast SNARE Spo20 fused to GFP, as PA sensor, and Lact-C2-GFP, the C2 domain of bovine lactadherin fused to GFP, as PS sensor (Várnai & Balla, 1998; Yeung et al, 2008; Kassas et al, 2017). The cells also expressed VWF-mRFP as WPB marker to allow for the detection of individual WPB–PM fusion events characterized by a collapse of the elongated WPB shape into a spherical object (Erent et al, 2007; Chehab et al, 2017; Mietkowska et al, 2019). As reported before (Erent et al, 2007; Mietkowska et al, 2019), most WPB–PM fusions were observed almost immediately after histamine addition, although the exocytosis events were not synchronous and the exact time needed for each vesicle to fuse differed, most likely depending on whether vesicles had been stored close to or at the PM in a fusion-competent state (Verhage & Sørensen, 2008; Chasserot-Golaz et al, 2010). With the exception of Lact-C2, none of the phospholipid sensors showed a significant signal on intracellular WPB in either resting or stimulated HUVECs, suggesting that neither $PI(4,5)P_2$ nor PA are particularly enriched in the membrane of WPB (for confocal images showing cytosolic WPB, see Fig S1). In contrast, all sensors stained the PM already in resting HUVEC revealing the expected PM localization of the target lipids.

When stimulated with histamine, PH-PLCδ1-YFP and to a somewhat lesser extent Spo20p3-GFP transiently accumulated at sites of WPB-PM fusion identified by the characteristic shape change of the elongated WPB (stills of live cell TIRF microscopy imaging are shown in Fig 1A and B, and those of live cell confocal microscopy in Fig S1A and B). Accumulation often occurred on one side of the fusion pore and with our TIRF settings was not observed in all fusion events, possibly because it was very transient or difficult to image above the general PM fluorescence. In contrast to the $PI(4,5)P_2$ and PA sensors, no obvious accumulation of the PS sensor Lact-C2-pEGFP was observed at histamine-evoked WPB-PM fusion sites (Fig 1C). Moreover, sensors for two other phosphoinositides, GFP-AKT-PH, the $PI(3,4,5)P_3$ binding domain of AKT fused to GFP, and mCherry-2XP4M-PI4P, a mCherry fusion of the PI(4)P binding domain of SidM, showed no specific enrichment at sites of WPB-PM fusion (Fig S1D and E). Fusion of secretory vesicle with the PM might transiently increase the sheer amount of membrane at the fusion sites. As this could bias the accumulation analysis of lipid markers, we performed control experiments with HUVECs expressing CAAX-GFP as a general PM marker. As shown in Fig 1D, histamine-induced WPB exocytosis and VWF secretion was not accompanied by an enrichment of CAAX-GFP at WPB–PM fusion sites. These data suggest that some [$PI(4,5)P_2$ and PA], but not all PM phospholipids transiently and specifically accumulate at the secretory organelle fusion sites in stimulated HUVEC.

### Depletion of $PI(4,5)P_2$ leads to a reduction in evoked WPB exocytosis

Next, we aimed to assess whether the enrichment at WPB–PM fusion sites reflects a functional involvement of the respective phospholipid. As $PI(4,5)P_2$ showed the stronger enrichment, we focused our functional analyses on this lipid. First, we used neomycin, a polycationic glycoside that binds to and masks $PI(4,5)P_2$ thereby, for example, inhibiting the insulin-stimulated exocytosis of

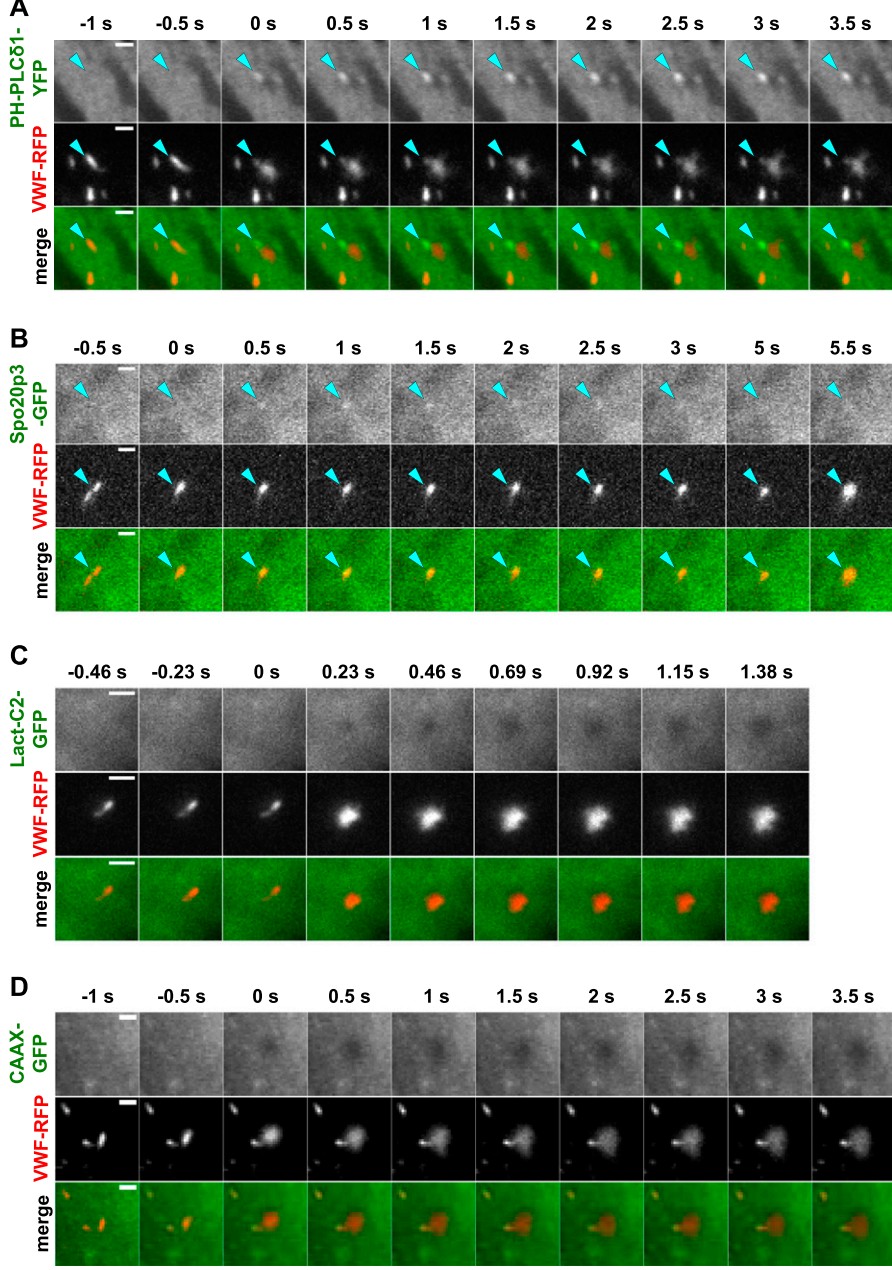

**Figure 1. PI(4,5)P$_2$ and phosphatidic acid but not PS accumulate at Weibel–Palade body (WPB)–plasma membrane fusion sites.**

**(A, B, C, D)** HUVECs expressing PH-PLCδ1-YFP (A), Spo20p3-GFP (B), Lact-C2-GFP (C), or CAAX-GFP (D) together with von-Willebrand factor-RFP as WPB marker were stimulated with 100 μM histamine and imaged by live-cell total internal reflection fluorescence microscopy. Still images show individual WPB undergoing exocytosis. Fusion occurred at t = 0 s. Note the transient enrichment of PH-PLCδ1-YFP and Spo20p3-GFP on the side of the actual fusion pore which appears darker in the total internal reflection fluorescence microscopy settings, most likely because the WPB membrane did not fully collapse into the plasma membrane during the time interval of the recordings. Scale bar = 2 μm.

Glut4 vesicles (James et al, 2004). Treatment of HUVECs with neomycin before histamine stimulation led to a significant reduction of evoked VWF release as compared with mock-treated cells (Fig 2). As PI(4,5)P$_2$ could also function as PLC substrate in the histamine receptor signaling cascade that ultimately leads to intracellular Ca$^{2+}$ elevation as a prerequisite for initiation of WPB exocytosis, we assessed whether neomycin also affects WPB exocytosis initiated by signaling cascades bypassing the histamine receptor. When ionomycin was used as a Ca$^{2+}$ ionophore that directly elevates intracellular Ca$^{2+}$, neomycin showed the same inhibitory effect on the acute VWF release as observed following histamine stimulation (Fig 2).

As these data suggest a functional involvement of PI(4,5)P$_2$ in WPB exocytosis, we next attempted to more specifically manipulate

PM PI(4,5)P$_2$ and analyze the consequences for histamine-evoked WPB exocytosis. Therefore, we adopted the rapamycin-inducible PI(4,5)P$_2$ depletion method that uses a type IV 5-phosphatase to convert PI(4,5)P$_2$ into PI4P (Varnai et al, 2006). Specifically, we used a dimerizer system comprising the phosphatase coupled to the FK506-binding protein (FKBP) and the FKBP-rapamycin binding domain of mTOR (FRB) coupled to a PM targeting palmitoylation signal (Fig 3A). HUVECs expressing PM-FRB-CFP and mRFP-FKPB-5-ptase showed a membrane distribution of PM-FRB-CFP and a cytosolic localization of mRFP-FKBP-5-ptase. Rapamycin treatment induced a significant translocation of mRFP-FKBP-5-ptase to the PM, which is not seen in mock-treated conditions. To verify that this rapamycin-induced PM recruitment of mRFP-FKBP-5-ptase caused

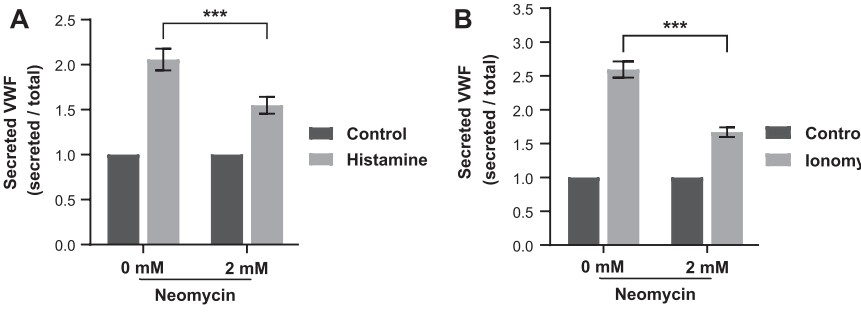

**Figure 2. Neomycin treatment inhibits Ca²⁺-evoked von-Willebrand factor (VWF) secretion.**
**(A)** HUVECs were treated with 2 mM neomycin for 20 min, incubated in agonist-free medium for 20 min, and then stimulated with 100 μM histamine for 20 min. VWF released into the cell culture supernatant was quantified by ELISA and secretion normalized to the total cellular VWF content (see the Materials and Methods section). n = 6, paired $t$ test, ***$P ≤ 0.001$. Bars represent mean ± SEM. **(B)** HUVECs were treated as in (A), but stimulated with 10 μM ionomycin instead of histamine. n = 10, paired $t$ test, ***$P ≤ 0.001$. Bars represent mean ± SEM. Note the marked reduction of evoked VWF secretion following neomycin treatment.

a PM PI(4,5)P₂ depletion, we cotransfected HUVEC with mRFP-FKPB-5-ptase, FRB-mRFP, and PH-PLCδ1-YFP as PI(4,5)P₂ sensor. As compared with the DMSO-treated control condition, where PH-PLCδ1-YFP shows a general PM distribution with some focal enrichments most likely reflecting PI(4,5)P₂-rich membrane protrusions and ruffles, rapamycin treatment induced a redistribution of PM-associated PH-PLCδ1-YFP to the cytoplasm (Fig 3B).

As a second control to verify the rapamycin-induced and type IV 5-ptase-dependent depletion of PM PI(4,5)P₂ in HUVEC, we analyzed the clathrin-dependent endocytosis of transferrin, which had been shown previously in other cells to be inhibited by PI(4,5)P₂ depletion (Kim et al, 2006; Varnai et al, 2006). In agreement with this, our data showed that transferrin uptake was profoundly inhibited in FRB-CFP and mRFP-FKPB-5-ptase expressing HUVEC after treatment with rapamycin (Fig S2A and B). Thus, rapamycin-induced membrane translocation of a type IV 5-ptase efficiently reduced PM PI(4,5)P₂ levels in HUVEC.

Using this system, we then analyzed whether PM PI(4,5)P₂ is functionally involved in Ca²⁺-dependent WPB exocytosis. HUVEC were transfected with FRB-CFP and either mRFP-FKPB-only or mRFP-FKPB-5-ptase and the effect of rapamycin-induced depletion of PM PI(4,5)P₂ on histamine-evoked WPB exocytosis and VWF secretion was analyzed by an anti-VWF antibody capture assay. This assay used DL-650–coupled anti-VWF antibodies, which were added to the medium and labeled sites of WPB exocytosis because of the extracellular emergence of VWF (Mietkowska et al, 2019). As shown in Fig 3C and D, the number of histamine-induced anti–VWF-DL650 spots representing individual WPB–PM fusion events was significantly reduced in mRFP-FKPB-5-ptase expressing and rapamycin treated HUVEC as compared with DMSO-treated control cells. In contrast, HUVEC transfected with mRFP-FKPB-only, that is, a construct devoid of the 5-ptase domain showed no change in the number of histamine-evoked fusion events when comparing DMSO- and rapamycin-treated conditions (Fig S2C). This is in line with the neomycin experiments and further suggests that PI(4,5)P₂ acts as a positive regulator of WPB exocytosis.

### The PIP5Kγ87 isoform is required for WPB exocytosis

We next assessed the role of PI(4,5)P₂ in evoked WPB exocytosis by targeting the major PI(4,5)P₂ generating enzyme, PI4P 5-kinase (PIP5K). As the PIP5K enzyme family consists of three isoforms, α, β, and γ (Doughman et al, 2003), we first examined which isoforms are expressed in HUVEC. Fig 4A shows that only PIP5Kγ is present at

significant levels. This is in contrast to the situation in HEK293T cells where PIP5Kα and PIP5Kβ are the most abundant isoforms (Fig 4A). Therefore, we focused on PIP5Kγ87 when analyzing the role of the PIP5K/PI(4,5)P₂ axis in WPB exocytosis.

Next, PIP5Kγ87 was depleted from HUVECs with specific siRNAs yielding a knockdown efficiency of ~80% without affecting the (low) levels of PIP5Kα protein (Fig 4B and C). Histamine-evoked WPB exocytosis was then analyzed in PIP5Kγ87-depleted HUVEC, both by quantifying the amount of VWF released into the culture medium and by determining the number of WPB-PM fusion events. Fig 4D shows that knockdown of PIP5Kγ87 caused a significant reduction in the amount of acutely secreted VWF in comparison with treatment with siRNA-control. Similarly, live cell imaging of histamine-evoked WPB-PM fusion events showed that the number of these fusion events was strongly decreased in PIP5Kγ87 depleted as compared with siRNA-control transfected HUVECs (Fig 4E). This result was corroborated by anti-VWF antibody capture assays revealing a significant decrease in the number of histamine-evoked anti-VWF-DL650 spots in PIP5Kγ87 depleted as compared with control cells (Fig 4F).

### PI(4,5)P₂ affects Ca²⁺ signaling and WPB exocytosis in histamine-stimulated HUVECs

Histamine stimulation of HUVECs elevates cytosolic Ca²⁺, which in turn causes acute WPB exocytosis to release VWF into the vasculature. To analyze whether this elevation of intracellular Ca²⁺ is compromised in cells showing reduced PM PI(4,5)P₂ levels following PIP5Kγ87 depletion, we loaded HUVECs with the green-fluorescent Ca²⁺ indicator Fluo-4-AM and the ratiometric Ca²⁺ dye Fura Red-AM and recorded [Ca²⁺]ᵢ changes in response to histamine. As shown in Fig 5A and B, knockdown of PIP5Kγ87 leads to a small reduction in the histamine-evoked [Ca²⁺]ᵢ elevation in comparison with that of control siRNA-treated cells. This suggests that the inhibitory effect of PIP5Kγ87 depletion on histamine-stimulated WPB exocytosis is mainly caused by an inhibition of WPB-PM docking and/or fusion events as a result of PM PI(4,5)P₂ depletion and not an altered Ca²⁺ signaling.

### PI(4,5)P₂ and PA function synergistically in histamine-evoked WPB exocytosis

PLD1, the enzyme generating PA, has been identified previously as a positive regulator of histamine-evoked WPB exocytosis (Disse et al,

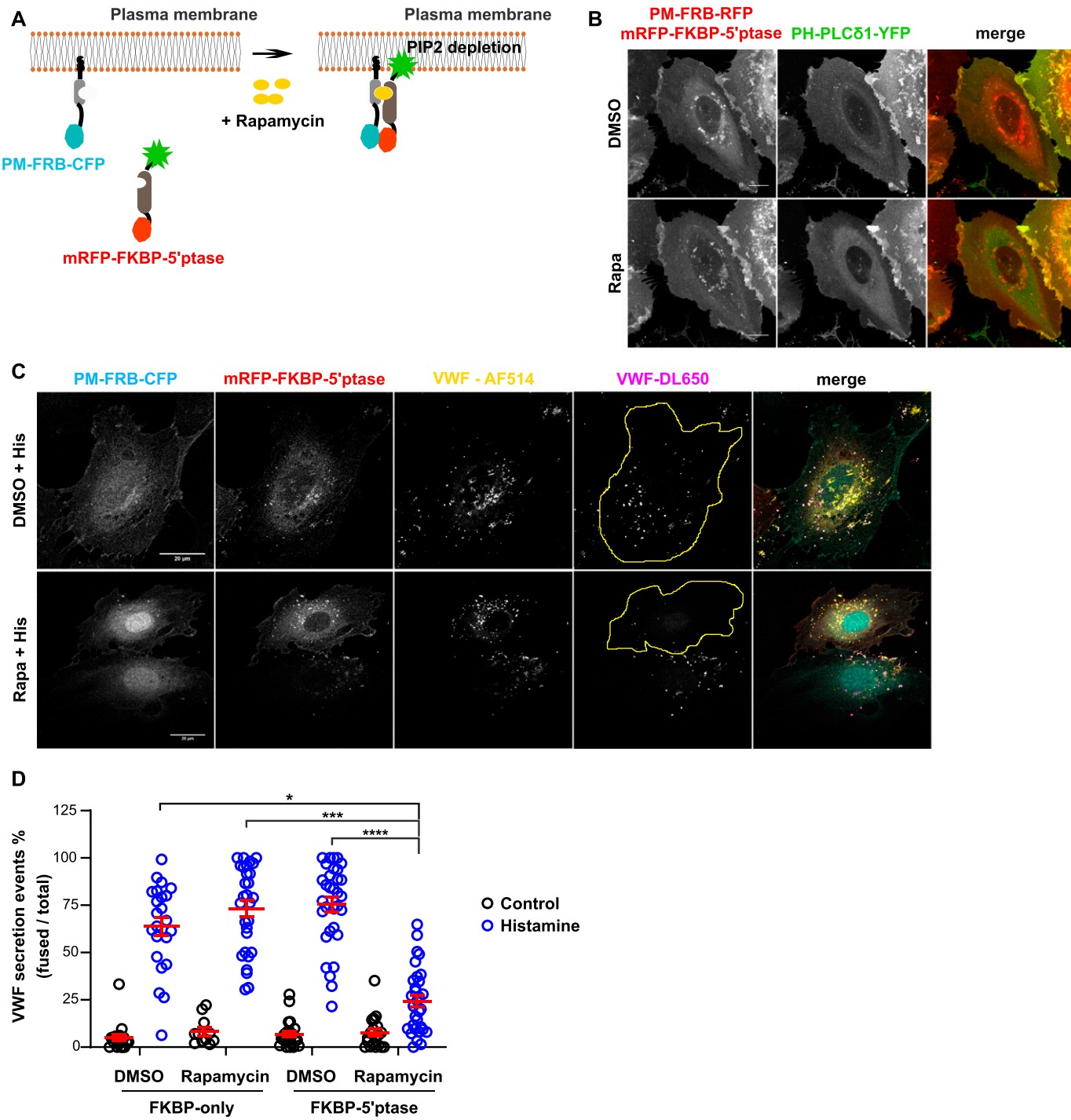

**Figure 3. Acute depletion of plasma membrane (PM) PI(4,5)P$_2$ inhibits histamine-evoked Weibel–Palade body (WPB) exocytosis.**
**(A)** Illustration depicting the rapamycin-inducible PI(4,5)P$_2$ depletion system. A 5-ptase fused to mRFP-FKBP is expressed together with PM-targeted FRB-CFP. Rapamycin treatment induces the interaction of mRFP-FKBP-5-ptase with PM-FRB-CFP, causing a PM translocation of the 5-ptase and a resulting reduction of PM PI(4,5)P$_2$. Adapted from Varnai et al (2006). **(B)** Representative maximum intensity projection images showing the PM localization of mRFP-FKBP-5-ptase and a PI(4,5)P$_2$ depletion upon rapamycin treatment. HUVECs were transfected with PM-FRB-RFP, mRFP-FKBP-5-ptase, and the PI(4,5)P$_2$ sensor PH-PLCδ1-YFP and live cell imaging was performed with time-lapse confocal microscopy, whereas rapamycin was added during acquisition. Translocation occurred between 3 and 7 min after rapamycin addition. In the flat HUVECs, this translocation is best seen in the rapamycin-induced increase of cytosolic PH-PLCδ1-YFP, which reflects itself in a more pronounced perinuclear fluorescence (thickest part of the cell containing most of the cytoplasm). Scale bar = 10 μM. **(C, D)** Representative confocal images and quantification of the effect of rapamycin-induced PM PI(4,5)P$_2$ depletion on WPB exocytosis. HUVECs were transfected with PM-FRB-CFP and mRFP–FKPB-only or mRFP-FKBP-5-ptase. 24 h post-transfection cells were treated with rapamycin for 3 min, stimulated with histamine for 15 min, and subjected to the anti–von-Willebrand factor (VWF) antibody capture assay as described in the Materials and Methods section. VWF-AF514 labels total VWF, whereas VWF-DL650 only labels the secreted VWF and thus identifies sites of WPB

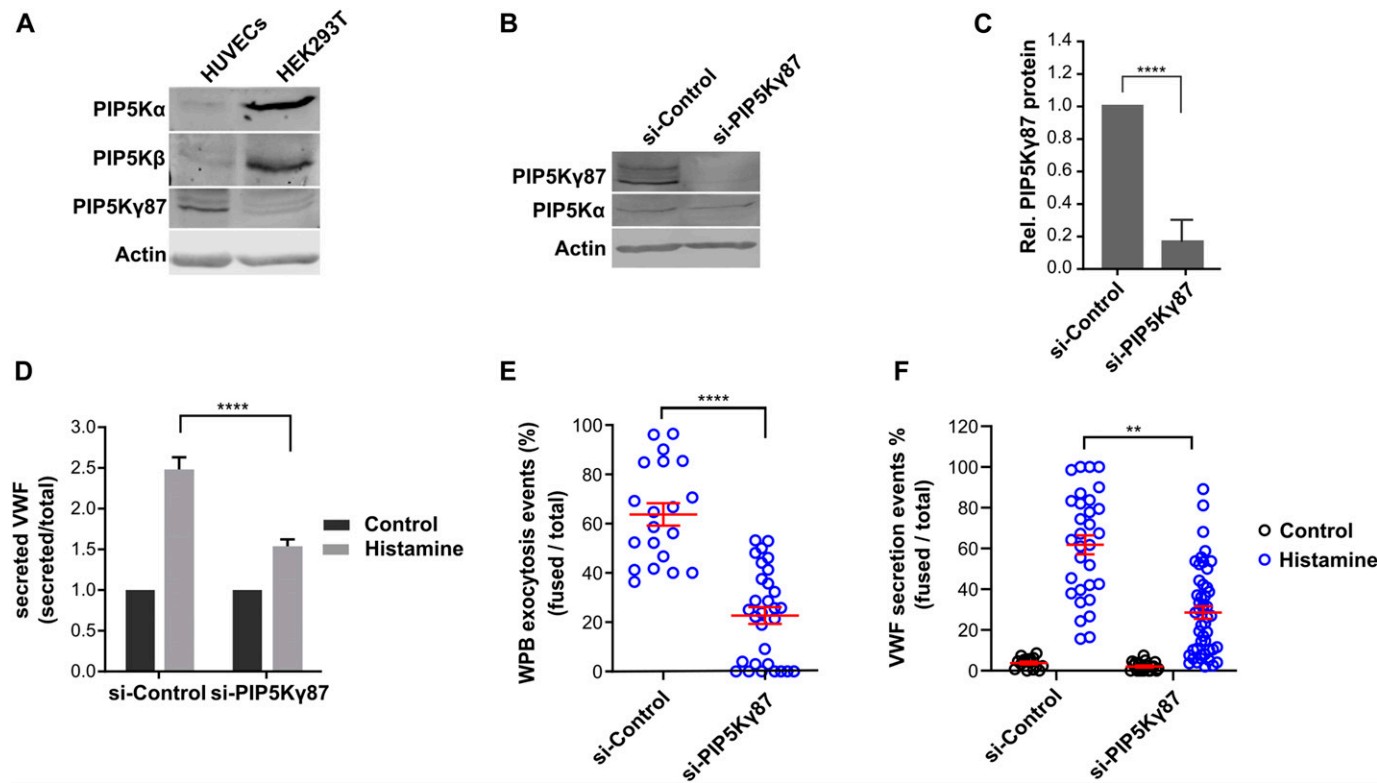

Figure 4. PIP5Kγ87 knockdown decreases histamine-evoked Weibel–Palade body (WPB) exocytosis.
(A) Expression of PIP5Kα, PIP5Kβ, and PIP5Kγ in HUVECs and HEK293T. HUVECs and HEK293T cell lysates were subjected to Western blot analysis using isoform-specific antibodies. Actin served as loading control. (B, C) PIP5Kγ knockdown. HUVECs were transfected with either siRNA-control or siRNA-PIP5Kγ87, and the knockdown efficiency was analyzed by Western blot of total cell lysates and quantified. (D) Quantification of von-Willebrand factor (VWF) secretion in PIP5Kγ87 depleted HUVECs. HUVECs were transfected as in (B), stimulated with histamine for 15 min, and the amount of VWF released into the culture supernatant was measured by ELISA. Data were analyzed by two-way ANOVA Tukey test of n = 5 independent experiments, ****$P < 0.0001$. Bars represent mean ± SEM. (E) siRNA-control and siRNA-PIP5Kγ87 knockdown cells were transfected with VWF-RFP and cultured on μ-slide eight well glass bottom dishes. Live cell imaging was then performed using time-lapse confocal microscopy and histamine was added during acquisition. The percentage of VWF secretion events was calculated as the ratio of fusion events per total number of WPB present in the respective cell at basal condition, that is, before addition of histamine. n = 3 independent experiments, Mann Whitney test, ****$P < 0.001$. Scatter plots represent mean ± SEM. (F) HUVECs were transfected as in (B), and VWF secretion was measured using the anti-VWF antibody capture assay. The percentage of VWF secretion was calculated by dividing the total number of secretion events visualized by antibody capture by the total number of WPB present in the respective cell. n = 3 independent experiments, Kruskal–Wallis test, ****$P < 0.001$. Scatter plots represent mean ± SEM.

2009) and PA has been shown to be a specific activator of PIP5Ks (Jenkins et al, 1994; Roach et al, 2012; Shulga et al, 2012). Thus, to address a possible functional link between PI(4,5)P$_2$ and PA in VWF secretion, we analyzed whether PA could function as an upstream activator of PIP5K and thus PI(4,5)P$_2$ production in the regulation of acute WPB exocytosis. Therefore, we generated a PIP5Kγ87 mutant, in which four residues shown previously to mediate PA binding, lysine 97, arginine 100, histidine 126, and histidine 127 (K97/R100/H126/H127 or KRHH) (Roach et al, 2012) were replaced by alanine thereby disrupting an interaction between PIP5Kγ87 and PA (Fig 6). If acting in a dominant-negative manner, this mutant should interfere with a PA-triggered elevation of PM PI(4,5)P$_2$ levels. Fig 6A shows that the KRHH-PIP5Kγ87 mutant, when expressed in a GFP-tagged form, has lost the ability to associate with the PM when

compared with wild-type-PIP5Kγ87 (WT-PIP5Kγ87-GFP). On the other hand, a kinase-dead mutant, in which D253 and R427 had been replaced by Asn and Gln, respectively, and which is characterized by compromised kinase activity but unperturbed PA binding (KD-PIP5Kγ87-GFP) (Coppolino et al, 2002; Roach et al, 2012; Nguyen et al, 2013) still localized to the PM. Moreover, whereas WT- and KD-PIP5Kγ87-GFP are recruited to WPB-PM fusion sites after histamine stimulation, KRHH-PIP5Kγ87-GFP fails to show this specific enrichment (Fig 6B).

To assess functional consequences of expressing the PA-binding mutant in HUVECs, we again analyzed transferrin uptake as a process known to be affected by PM PI(4,5)P$_2$ depletion (see above). KRHH-PIP5Kγ87-GFP expressing HUVECs indeed showed a significant decrease in their ability to internalize transferrin as compared with

exocytosis. Circumferences of cells positive for both PM-FRB-CFP and mRFP-FKBP-5-ptase are highlighted in yellow. (D) Shows a quantification of the anti-VWF antibody capture data from n = 3 independent experiments. Scatter plots represent mean ± SEM. Representative confocal images of the controls including mRFP-FKPB-only are shown in the Fig S2. Scale bar = 20 μM. Significance was tested with Kruskal–Wallis test, ***$P < 0.0001$.

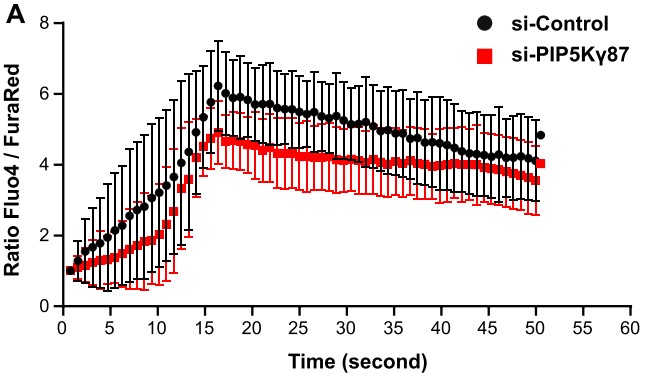

**Figure 5. PIP5Kγ87 knockdown partially inhibits intracellular calcium elevation in response to histamine stimulation.**
PIP5Kγ87 control and knockdown cells were loaded with 2 μM of each, Fluo-4-AM and Fura Red-AM, and emission signals were recorded using live cell imaging by confocal microscopy. **(A, B)** The actual recordings for a representative cell are shown in (A), whereas (B) gives a quantification of the Ca$^{2+}$ signals observed in n = 5 independent experiments with a total of 294 cells. Bars represent mean ± SEM. Significance was tested with Mann–Whitney test, ****$P$ < 0.0001.

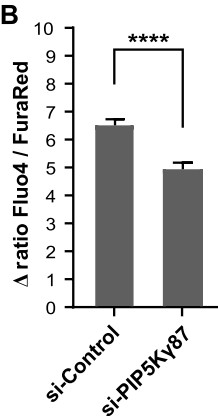

cells expressing WT- or KD-PIP5Kγ87-GFP (Fig S3). This suggests that in contrast to KD-PIP5Kγ87-GFP, KRHH-PIP5Kγ87-GFP acts in a dominant-negative manner, most likely by interfering with a PA-mediated recruitment/activation of PIP5Kγ87 and thus PM PI(4,5)P$_2$ production. Next, we examined whether WPB exocytosis is also affected by KRHH-PIP5Kγ87-GFP expression. As before, histamine-evoked WPB-PM fusion events were visualized and quantified by anti-VWF antibody capture in HUVECs transfected with either WT-PIP5Kγ87-GFP, KRHH-PIP5Kγ87-GFP or KD-PIP5Kγ87-GFP, respectively. As shown in Fig 6C, KRHH-PIP5Kγ87-GFP expression caused a significant reduction in histamine-evoked exocytotic fusion events, whereas expression of either WT- or KD-PIP5Kγ87-GFP did not result in major changes in the number of fusions when compared with mock-transfected HUVECs.

### PIP5Kγ87 is required for PLD1 activation in HUVECs

The above data suggest that PA-mediated recruitment of PIP5Kγ87 to the PM (and presumably WPB-PM fusion sites) results in local PI(4,5)P$_2$ production which supports WPB exocytosis. PM PI(4,5)P$_2$, on the other hand, is a known activator of the PA-generating enzyme PLD1 suggesting a possible mutual activation loop involving PIP5Kγ87 and PLD1 and their reaction products PI(4,5)P$_2$ and PA. To address this possible link we analyzed whether elevated PI(4,5)P$_2$ levels can also activate PLD1 and thus PA production in secretagogue-treated HUVECs. Therefore, we depleted HUVECs of PIP5Kγ87 and then determined PLD1 activity in control versus histamine stimulation conditions. Fig 7A shows that PIP5Kγ87 knockdown significantly dampened the histamine-induced increase in PLD1 activity which previously had been reported to elicit WPB exocytosis (Disse et al, 2009; Huang et al, 2012). In line with all previous data, this suggests that a positive feed-forward loop involving PIP5Kγ87 and PLD1 and elevated PM PI(4,5)P$_2$ and PA levels operates in histamine-stimulated HUVECs in the course of WPB exocytosis (Fig 7B).

## Discussion

The exocytosis of WPB allows endothelial cells to rapidly respond to vascular injury by secretion of highly multimeric VWF. This process has to be tightly regulated to prevent excessive VWF secretion and thus coagulation under resting conditions. Regulation of WPB exocytosis is achieved through the involvement of second messenger systems (Ca$^{2+}$, cAMP) that transmit signals to the machinery executing WPB docking and fusion with the PM. Several proteins of this machinery have been identified and many of these proteins are known to bind PM lipids that could serve recruitment functions and/or regulate protein activities. However, a role of lipids in the regulation of WPB exocytosis had so far remained elusive. Here, we exploited the unique shape and dynamics of WPB to visualize individual secretory events in live cells with high spatiotemporal resolution and related these exocytotic fusions to PM lipid distributions recorded by using fluorescently labeled phospholipid binding domains. Thereby, we identified two phospholipids, PI(4,5)P$_2$ and PA, that become transiently enriched at WPB-PM fusion sites and that, together with their synthesizing enzymes PIP5Kγ87 and PLD1, most likely act in a stimulatory feed-forward loop to support WPB exocytosis and thus VWF secretion.

A contribution of PI(4,5)P$_2$ had been reported before in other types of exocytotic fusion events where the lipid appears to participate in protein-lipid interactions and protein activation at a pre-fusion stage (Holz, 2006; Martin, 2015; Lauwers et al, 2016; Gawden-Bone et al, 2018). In neurons and neuroendocrine cells, the two best studied cell types in this respect, PI(4,5)P$_2$ has been shown to interact with components of the protein machinery involved in docking, priming and PM fusion of synaptic vesicles and dense core granules, for example, to facilitate the ability of Munc13 to induce SNARE complex formation, to increase syntaxin clustering and to regulate the membrane interaction of synaptotagmin-1, CAPS, or Doc2β and thus their role in vesicle docking and fusion (Kabachinski et al, 2014; Park et al, 2015; Petrie et al, 2016; Walter et al, 2017; Bradberry et al, 2019). By interacting with proteins of the (clathrin-mediated) endocytosis machinery, PI(4,5)P$_2$ can also function in membrane/protein internalization and probably contributes to the maintenance of a homeostatic balance between PM expansion by exocytosis and compensatory retrieval by endocytosis (Mettlen et al, 2018). Most likely, this is also of relevance in WPB exocytosis where clathrin- and dynamin-dependent membrane retrieval appears to occur at sites of WPB-PM fusion (Stevenson et al, 2017). In addition to their role in membrane trafficking, PI(4,5)P$_2$ rich PM membrane

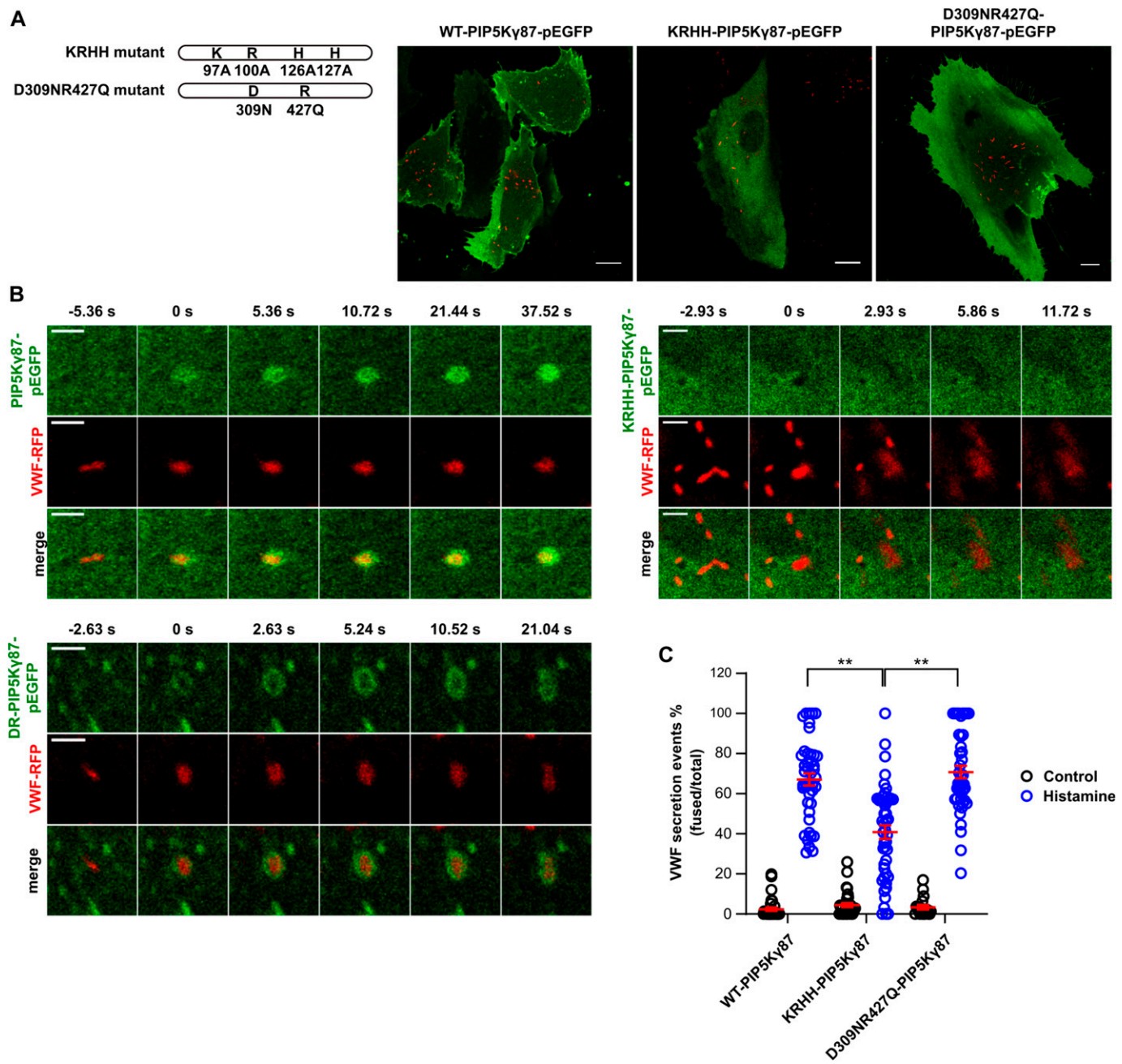

**Figure 6. The KRHH-PIP5Kγ87 mutant which is defective in phosphatidic acid binding is cytosolic and causes reduction of histamine-evoked Weibel–Palade body (WPB) exocytosis.**
**(A)** Left, illustration highlighting the point mutations introduced in PIP5Kγ87. Right, localization of the different PIP5Kγ87 derivatives. HUVECs were transfected with either WT-PIP5Kγ87-GFP, KRHH-PIP5Kγ87-GFP, or KD-PIP5Kγ87-GFP and von-Willebrand factor (VWF)-RFP. Cells were fixed 15 h post-transfection, and the localization of the PIP5Kγ87 constructs was recorded by confocal microscopy. Scale bar = 10 μM. **(B)** HUVECs were transfected as in (A), and confocal live cell imaging was performed after addition of histamine. Still images show individual WPB undergoing exocytosis at t = 0 s. **(C)** Expression of KRHH-PIP5Kγ87-GFP interferes with histamine-evoked WPB exocytosis. Anti-VWF antibody capture assay of cells expressing WT-PIP5Kγ87-GFP, KRHH-PIP5Kγ87-GFP, or KD-PIP5Kγ87-GFP, respectively, in control or histamine-stimulated conditions. The number of fusion events and the total number of WPB were quantified using ImageJ and the percentage of WPB exocytosis events was calculated as the ratio of fusion events per total number of VWF-positive WPB. Data of n = 3 independent experiments were analyzed by Kruskal–Wallis test, **$P < 0.01$. Scatter plots represent mean ± SEM.

domains represent sites for interactions with the underlying cortical actin cytoskeleton, again mediated by a subset of PI(4,5) P$_2$-binding proteins (Zhang et al, 2012; Senju & Lappalainen, 2019).

Such a link to the submembranous actin is possibly also relevant in WPB exocytosis as a subset of fusion events is accompanied by the post-fusion acquisition of an actin coat or ring that could aid

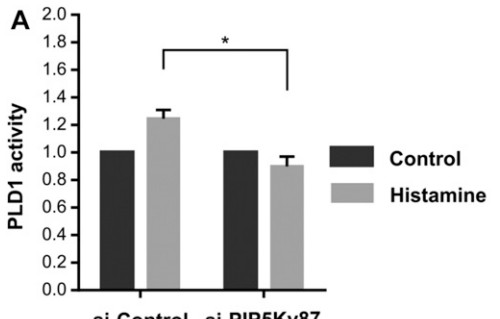

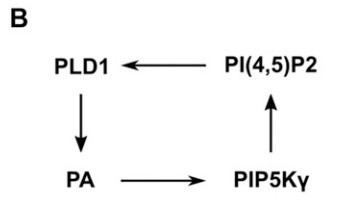

**Figure 7. PIP5Kγ87 depletion inhibits histamine-evoked PLD1 activation.**
**(A)** HUVECs transfected with either siRNA-control or siRNA-PIP5Kγ87 were mock-treated (control) or treated with histamine (100 mM) for 20 min. PLD1 activity was then measured as described in the Materials and Methods section. Data from n = 3 independent experiments, statistical analysis used two-way ANOVA, *$P < 0.05$. Bars represent mean ± SEM. **(B)** Feed-forward model in which PI(4,5)P$_2$ and phosphatidic acid regulate each other through recruitment/activation of the respective enzymes.

the expulsion of highly multimeric VWF and/or could be involved in endocytic membrane retrieval (Nightingale et al, 2018, 2011; Mietkowska et al, 2019). The actin nucleation factor(s) most likely regulating this actin assembly at post-fusion WPB have not been identified but, in line with the properties of many of the known nucleation (promoting) factors, they possibly require elevated PI(4,5)P$_2$ levels for PM recruitment. Thus, the PI(4,5)P$_2$ accumulations observed here at WPB–PM fusion sites might serve different functions. In addition to supporting WPB exocytosis, they could also provide signals for actin coat formation and/or endocytic membrane retrieval; future experiments will have to address this issue.

Several phospholipid/PI(4,5)P$_2$-binding proteins shown to participate in regulated exocytosis in other cells have also been identified as positive regulators of WPB exocytosis. They include the synaptotagmin-like protein Slp4-a, which is recruited to WPB by Rab27a binding, the tethering factor Munc13-4 and the WPB-associated synaptotagmin-5 which is likely to function as an important Ca$^{2+}$ sensor (Bierings et al, 2012; Chehab et al, 2017; Lenzi et al, 2019). However, the relevance of phospholipid/PI(4,5)P$_2$ binding for the function of these proteins in evoked WPB exocytosis has not been analyzed. Possibly, it involves providing a link between cortical WPB and PI(4,5)P$_2$ rich sites in the PM, which could be regulated by Ca$^{2+}$ and/or the relative PI(4,5)P$_2$ levels. At least for Munc13-4, we could show in earlier work that histamine stimulation triggers the formation of Munc13-4–rich foci at WPB–PM fusion sites, and that these foci are not observed for a Munc13-4–mutant protein lacking the C-terminal C2B domain which binds to acidic phospholipids, including PI(4,5)P$_2$ (Chehab et al, 2017). In addition to directly binding to PM PI(4,5)P$_2$, these factors could also interact with other PM-localized PI(4,5)P$_2$-binding proteins such as syntaxins or the AnxA2–S100A10 complex, which can cluster PI(4,5)P$_2$ into larger domains and has been shown to directly associate with WPB-bound Munc13-4 (Chehab et al, 2017). Lipid binding affinities and spatial as well as temporal constraints probably dictate the order of PI(4,5)P$_2$–protein interactions in the course of Ca$^{2+}$-regulated WPB exocytosis and subsequent compensatory endocytosis.

The accumulation of PI(4,5)P$_2$ at WPB–PM fusion sites seen in our TIRF microscopy settings most likely reflects a substantial enrichment of this lipid at sites of exocytosis. Indeed, whereas the average concentration of PI(4,5)P$_2$ in the inner PM leaflet is around 2 mol% it has been estimated to increase to more than 50 mol% in PI(4,5)P$_2$ rich microdomains observed in PM sheets of neuroendocrine PC12 cells

(van den Bogaart et al, 2011). PI(4,5)P$_2$ enriched microdomains have also been visualized in live PC12 cells; here, Ca$^{2+}$-evoked dense core vesicle exocytosis preferentially occurred at such sites and was facilitated by an activation of CAPS and a recruitment of Munc13, both dependent on PI(4,5)P$_2$ (Kabachinski et al, 2014). Interestingly, our live cell analysis revealed that only a subset of WPB-PM fusion sites show a marked enrichment of the PI(4,5)P$_2$ sensor PH-PLCδ1-YFP. This could reflect the transient nature of the actual docking/fusion event characterized by the PI(4,5)P$_2$ dependent recruitment of docking and fusion factors such as Munc13-4. However, it is also possible that the different WPB-PM fusion and post-fusion structures observed before differ in their requirement for PI(4,5)P$_2$. For instance, exocytotic events characterized by the post-fusion assembly of an actin ring or coat around the not fully collapsed WPB membrane, which aids large cargo expulsion, are likely to occur at PI(4,5)P$_2$–rich domains because this could facilitate the recruitment of actin polymerization factors (Nightingale et al, 2018; Mietkowska et al, 2019).

We also observed an enrichment of PA at WPB–PM fusion sites in histamine-evoked VWF exocytosis. As shown in other systems, an accumulation of PA in the inner cytosolic leaflet of vesicle-PM contact sites facilitates membrane curvatures that support fusion (Bullen & Soldati-Favre, 2016; Tanguy et al, 2018). In addition, PA could act by recruiting proteins mediating evoked WPB exocytosis. As an acidic phospholipid, PA interacts with several of the PI(4,5)P$_2$ binding docking and fusion factors discussed above and could thus participate in their recruitment. Moreover, it binds to and recruits PIP5Kγ to WPB–PM contact/fusion sites and can thereby trigger a local enrichment of PI(4,5)P$_2$. The latter can be inferred because overexpression of the KRHH-PIP5Kγ87 mutant which is defective in PA binding has a dominant-negative effect on evoked WPB exocytosis. Hence, we assume that PA-mediated membrane recruitment of PIP5Kγ87 generates local PI(4,5)P$_2$ accumulations that promote WPB–PM docking and/or fusion. However, our experiments do not rule out the possibility that other mechanisms contribute to the formation of PI(4,5)P$_2$ rich domains at WPB-PM fusion sites, for example, the local inhibition of a PI(4,5)P$_2$ phosphatase or lateral lipid segregation in the plane of the PM mediated by proteins such as AnxA2. Interestingly, we could show that PIP5Kγ87 depletion also abrogates the histamine-induced increase in PLD1 activity. As PI(4,5)P$_2$ is a known activator of PLD1 and PLD1 has been shown to promote histamine-evoked WPB exocytosis (Disse et al, 2009), we assume that PIP5Kγ87 and PLD1 are part of a feed-forward loop generating elevated PA and PI(4,5)P$_2$

levels at PM sites where WPB dock and fuse. Future experiments have to establish the upstream events that initiate this feed-forward loop after endothelial stimulation and that could involve the histamine-induced activation of Arf6 as a regulator of PLD1 and WPB exocytosis (Biesemann et al, 2017).

# Materials and Methods

## Reagents and antibodies

Histamine (H0537) and rapamycin (R8781) were purchased from Sigma-Aldrich. Ionomycin (Cay11932-1) was obtained from Biomol (Biomol GmbH). siRNAs against PIP5Kγ87 were designed and purchased from Dharmacon (Dharmacon Inc.) and AllStars Negative Control siRNA (102781) was obtained from QIAGEN. All antibodies against VWF were purchased from Dako. Rabbit polyclonal anti-VWF-Dylight650 antibodies were generated by coupling the dye to rabbit anti-VWF antibodies (Dako) using the DyLight antibody labeling kit (Thermo Fisher Scientific). Antibody against PIP5Kγ was a gift from Kun Ling (Mayo Clinic Cancer Center). Alexa Fluor 488–conjugated Transferrin (human) (T13342) and FM 4-64 dye (N-(3-Triethylammoniumpropyl)-4-(6-(4-(Diethylamino) Phenyl) Hexatrienyl) Pyridinium Dibromide) (T13320) were purchased from Thermo Fisher Scientific. Fluo-4-AM Ultrapure Grade (20551) and Fura Red AM (21048) were obtained from ATT Bioquest.

## Expression constructs

The VWF-mRFP plasmid was kindly provided by Tom Carter (St George's University of London). GFP-PIP5Kγ87 was purchased from Addgene (Plasmid No: 22300). GFP-PIP5Kγ87-K97A/R100A/H126A/H127A (KRHH) and GFP-PIP5Kγ87-D309N/R427Q (DR) mutants were generated using the Q5 Site-Directed Mutagenesis Kit (New England Biolabs Inc.) according to the manufacturer's instruction. The following primers were used to introduce the respective mutations: gaagcCGACGTGCTCATGCAGGAC as KR forward primer, gggcgcGGAGCTCAGGTGGCCCAC as KR reverse primer; GACCCCTGCTgctgcCTACAATGACTTTCG as HH forward primer, AGGTTGCTCCCTTCACTG as HH reverse primer; CAAGATCATGaACTACAGCCTG as D309N forward primer, AAACTTTCCAGGACCAGG as D309N reverse primer; TATGCCGAGCagTTTTTCAAGTTCATGAGCAACACGG as R427Q forward primer, GAAGCTGGGGCGGTGGAC as R427Q reverse primer. PH-PLCδ1-YFP was obtained by subcloning the PH-PLCδ1 domain into pEYFP-N1 as described for PLC1-PH-GFP (Varnai et al, 2006). PH-PLCδ1-GFP (plasmid no. 51407) and PH-PLCD1-R40L-GFP (plasmid no. 51408) were obtained from Addgene (kindly deposited by Tamas Balla) and the PH domain inserts subcloned into pEGFP-N1 (Clontech) via EcoRI/BamHI restriction sites. Lact-C2-pEGFP was a gift from Sergio Grinstein (plasmid no. 22852; Addgene) and Spo20p3-GFP was kindly provided by Nicolas Vitale (University of Strasbourg). PM-FRB-CFP (Plasmid no. 67517), mRFP-FKBP (mRFP-FKBP only) (Plasmid no. 67514) and mRFP-FKBP12-5'ptase-domain (mRFP-FKBP-5'ptase) (Plasmid no. 67516) were generated by Tamas Balla (Varnai et al, 2006) and purchased from Addgene (kindly deposited by Tamas Balla). GFP-AKT-PH (plasmid no. 21218; Addgene) was a gift from

Andreas Püschel (University of Münster). mCherry-2XP4M-PI4P (Hammond et al, 2014) was kindly provided by Nina Criado Santos (University of Geneva).

## Cell culture and transfection

Human umbilical vein endothelial cells (HUVECs) were purchased from Promocell (C-12203) and cultured on Corning CellBIND plates at 37°C and 5% $CO_2$ in 1:1 mixed medium of ECGM2 (Promocell) and M199 (Biochrom), which was supplemented with 10% FCS (Sigma-Aldrich), 30 μg/ml gentamycin, and 15 ng/ml amphotericin. All plasmid and siRNA transfections were performed by electroporation using the Amaxa HUVEC Nucleofector Kit (Lonza) according to the manufacturer's instructions.

To efficiently knockdown PIP5Kγ, HUVECs were subjected to two rounds of the respective siRNA transfection. At each round, $1 \times 10^6$ cells were transfected with 100 nM of either siRNA control or siRNA-PIP5Kγ and cells were cultivated for 48 h after the first and 24 h after the second transfection. Knockdown efficiency was confirmed by Western blot analysis.

## Western blot analysis

HUVECs were harvested in cold lysis buffer (50 mM Tris, pH 7.4, 150 mM NaCl, 1 mM EDTA, 1% NP-40, 0.5% sodium deoxycholate, and 1× Complete EDTA-free Proteinase Inhibitor Cocktail [PI]) and lysed on ice for 30 min. After clearance by centrifugation (15,000g, 15 min, 4°C), the protein concentration of the cell lysates was measured using the Pierce 660 nm protein assay reagent (Thermo Fisher Scientific). Equal amounts of total protein in Laemmli sample buffer were subjected to SDS–PAGE and transferred to nitrocellulose membranes (GE Healthcare Life Science). Membranes were blocked with 5% nonfat milk in TBS containing 0.1% Tween-20 (TBST) (1 h, RT) and incubated with the corresponding primary antibodies (overnight, 4°C). After washing and treatment with IR dye–conjugated secondary antibodies (1 h, RT), bound IR dyes were visualized using the Odyssey Infrared Imaging System (Li-COR Bioscience).

## VWF secretion assay

The stimulated release of VWF into the cell culture medium was quantified by an ELISA-based assay (Chehab et al, 2017). Cells were seeded in triplicate on collagen-coated 24-well plates, grown to confluence, and starved overnight in starvation medium (M199 + antibiotic + 1.5% BSA). Starvation was omitted in the neomycin experiments. The medium was replaced by 200 μl of fresh medium, and cells were incubated for 20 min. The supernatant was then collected (basal condition) and replaced by 200 μl of histamine or ionomycin-containing medium. After another incubation for 20 min, the supernatants were again collected (stimulated condition), and the remaining cells were lysed in 200 μl of 0.1% Triton X-100–containing medium, followed by incubation for 30 min on ice. The amount of VWF in the supernatants (basal, stimulated) and the total lysate was then determined by ELISA (Disse et al, 2009), and the percentage of secreted VWF was expressed as VWF released in stimulation condition per total amount of VWF (sum of VWF measured in all three conditions).

## Anti-VWF antibody capture assay

Evoked WPB exocytosis was visualized microscopically by an anti-VWF antibody capture assay (Knop et al, 2004). Briefly, live cells grown at 37°C and 5% $CO_2$ were first incubated in rabbit anti-vWF (1:400; Dako)–containing medium for 20 min to saturate nonspecific antibody binding sites. Cells were then washed twice with warm PBS, new medium containing rabbit–anti–VWF-DyLight-650 conjugated antibodies was added and cells were incubated for 3 min. To induce VWF secretion, either histamine (100 $\mu M$) or ionomycin (10 $\mu M$) were added and incubation was continued for 15 min. The remaining steps were performed at RT and 3% BSA-containing PBS was used as buffer for blocking steps or to dilute antibodies. Cells were washed thrice, fixed in 4% PFA for 10 min, permeabilized in PBS containing 0.1% Triton X-100 for 2 min, washed thrice, and incubated with blocking buffer for 45 min. Mouse anti-VWF antibodies (clone F8/68, 1:400; Dako) were then added to label intracellular VWF and incubation was continued for 1 h at RT (or overnight at 4°C). Cells were then washed three times and incubated with goat antimouse AF514-conjugated antibodies (1:2,000) as secondary antibodies. Finally, cells were washed three times and mounted using mounting media.

## Quantification of WPB–PM fusion events

WPB exocytosis events were recorded using time-lapse confocal microscopy of VWF-RFP expressing HUVECs, whereas histamine was added within the first 10 frames of acquisition process. The total number of WPB before histamine addition and number of WPB exocytosis events identified by the characteristic shape change were counted manually in ImageJ. Final data are presented as a percentage of WPB exocytosis events per total number of WPB in the respective cell.

In the anti-VWF antibody capture assay, Z-stack data were acquired by confocal microscopy and the number of both total VWF-positive WPB (labeled with AF514) and VWF secretion events (labeled with DL-650) were counted in ImageJ. Data are presented as ratio between the number of VWF secretions events per total number of WPB in the respective cells (% of VWF-DL650/VWF-AF514).

## Transferrin uptake assay

HUVECs were seeded on 24-well plates and cultivated to 80% confluency. Old medium was replaced by fresh medium containing AF-488–conjugated transferrin and cells were incubated for 5 min at 4°C. Cells were then washed thrice with warm medium and incubated in fresh medium for an additional 10 min at 4°C. Transferrin-containing HUVECs were then washed once and fixed in 4% PFA for 10 min at RT. Finally, the cells were washed thrice with PBS and mounted using mounting medium.

## Confocal and TIRF microscopy

Cells were grown on collagen-coated eight-well glass bottom $\mu$-slides (ibidi) for live cell imaging or glass coverslips for immunocytochemistry. Live cell imaging was carried out in 25 mM Hepes-containing mixed HUVEC medium at 37°C and WPB exocytosis was stimulated with either histamine or ionomycin. The data were recorded using confocal (LSM 780, LSM 800; Zeiss) and inverted TIRF microscopes (IX83; Olympus).

## PLD activity assay

PLD1 activity was measured using the Amplex Red Phospholipase D Assay Kit (A12219; Thermo Fisher Scientific). In brief, HUVECs were cultured on 100-mm dishes until fully confluent. Cells were washed twice with cold PBS and placed on ice. To prepare cell lysates for the assay, cells were harvested in PI-containing 50 mM Tris–HCl buffer (pH 8.0) and subjected to three freeze–thaw cycles using liquid nitrogen. Lysates were cleared by centrifugation at 15,000g for 5 min at 4°C, protein in the supernatant was quantified and the PLD activity assay was then performed according to the assay kit protocol.

## Calcium measurements

The calcium indicator Fluo-4-AM was used together with the ratiometric $Ca^{2+}$ dye Fura Red-AM, both at final concentrations of 2 $\mu M$. To simultaneously visualize WPB and WPB exocytosis, HUVECs were transfected with VWF-RFP and cultivated on collagen-coated eight-well glass bottom $\mu$-slides for 20 h. Subsequently, Fluo-4-AM and Fura Red-AM, diluted in either 100 $\mu M$ EGTA- or 2.5 mM $Ca^{2+}$-containing Tyrode's buffer (140 mM NaCl, 5 mM KCl, 1 mM $MgCl_2$, 10 mM glucose, and 10 mM Hepes, pH 7.4), was added to the cells, followed by incubation for 20 min at RT. The cells were then washed twice with PBS and fresh Tyrode's buffer was added. After 5 min of incubation at 37°C, Fluo-4 and Fura Red-AM emission signals were recorded by confocal live cell imaging (LSM780).

## Statistical analysis

Statistical analyses were performed with GraphPad Prism 8. Quantification of Western blot data and percentage of WPB fusion events in live cells used a Mann–Whitney $t$ test and the anti-VWF antibody capture assay data were analyzed using the Kruskal–Wallis test.

# Supplementary Information

# Acknowledgements

We thank Wolf Almers (Vollum Institute, Portland, OR, USA) for advice on the TIRF microscopy experiments (in part carried out during a sabbatical stay of V Gerke) and general discussion. This work was supported by grants from the German Research Council (DFG Ge514/6-3 and SFB1348/A04). TTN Nguyen is supported by the graduate school of the Cells-in-Motion Cluster of Excellence (EXC 1003 – CiM), University of Münster, Germany.

## Author Contribution

TTN Nguyen: conceptualization, data curation, formal analysis, investigation, and writing—original draft, review, and editing.

SN Koerdt: conceptualization, data curation, formal analysis, investigation, and writing—original draft, review, and editing.

V Gerke: conceptualization, data curation, supervision, funding acquisition, and writing—original draft, review, and editing.

## Conflict of Interest Statement

The authors declare that they have no conflict of interest.

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
