## [Reviewer comments · Life Science Alliance]

Plasma membrane phosphatidylinositol (4,5)-bisphosphate promotes Weibel-Palade body exocytosis

Tu Thi Ngoc Nguyen, Sophia Koerdt, Volker Gerke

DOI: 10.26508/lsa.202000788

Corresponding author(s): Prof. Volker Gerke (University of Muenster)

Review timeline:

Submission Date:	2020-05-20
Editorial Decision:	2020-06-16
Revision Received:	2020-08-04
Editorial Decision:	2020-08-12
Revision Received:	2020-08-13
Accepted:	2020-08-14

Scientific Editor: Shachi Bhatt

Transaction Report:

No Peer Review Process File is available with this article, as the authors have chosen not to make the review process public in this case.

Re: Life Science Alliance manuscript #LSA-2020-00788-T

Prof. Volker Gerke
University of Muenster
Institute of Med. Biochemistry
Von-Esmarch-Str. 56
Muenster D-48149
Germany

Dear Dr. Gerke,

Thank you for submitting your manuscript entitled "Plasma membrane phosphatidylinositol (4,5)-biphosphate promotes Weibel-Palade body exocytosis" to Life Science Alliance. The manuscript was assessed by expert reviewers, whose comments are appended to this letter.

As you will see, the reviewers appreciate the interesting findings presented in your manuscript and the excellent quality of your data. Given the overall high level of interest in your study, we would like to invite you to submit a revised version of the manuscript that addresses the reviewers' comments. In particular, we feel that the request from reviewer 2 to show individual data points in your graphs should be addressed. For reviewer 1, while some comments may be outside the scope of the current study, we do not rule out that you may have data in-hand that could be added to the manuscript to address some of these points.

When submitting the revision, please include a letter addressing the reviewers' comments point by point. In our view these revisions should typically be achievable in around 3 months. However, we are aware that many laboratories cannot function fully during the current COVID-19/SARS-CoV-2 pandemic and therefore encourage you to take the time necessary to revise the manuscript to the extent requested above. We will extend our 'scooping protection policy' to the full revision period required. If you do see another paper with related content published elsewhere, nonetheless contact me immediately so that we can discuss the best way to proceed.

Thank you for this interesting contribution to Life Science Alliance. We are looking

forward to receiving your revised manuscript.

Sincerely,

Reilly Lorenz
Editorial Office Life Science Alliance
Meyerhofstr. 1
69117 Heidelberg, Germany
t +49 6221 8891 414
e contact@life-science-alliance.org
www.life-science-alliance.org

B. MANUSCRIPT ORGANIZATION AND FORMATTING:

RE: Life Science Alliance Manuscript #LSA-2020-00788-TR

Prof. Volker Gerke
University of Muenster
Institute of Med. Biochemistry
Von-Esmarch-Str. 56
Muenster D-48149
Germany

Dear Dr. Gerke,

Thank you for submitting your revised manuscript entitled "Plasma membrane phosphatidylinositol (4,5)-biphosphate promotes Weibel-Palade body exocytosis". We would be happy to publish your paper in Life Science Alliance pending final revisions necessary to meet our formatting guidelines.

Along with the points listed below, please also make the following edits to the manuscript text to comply with the formatting and editorial guidelines at LSA,

- please upload your supplementary figures as single files; these will be displayed in-line in the HTML version of your paper, so please provide them as single page files (figure S1 currently spans over two pages)
- please add a callout for Figure 1A & 1B and Figure S2C
- please add your supplementary figure legends to the main manuscript text
- please use the [10 author names, et al.] format in your references (i.e. limit the author names to the first 10)

A. FINAL FILES:

-- Summary blurb (enter in submission system): A short text summarizing in a single sentence the study (max. 200 characters including spaces). This text is used in conjunction with the titles of papers, hence should be informative and complementary to the title. It should describe the context and significance of the findings for a general readership; it should be written in the present tense and refer to the work in the third

person. Author names should not be mentioned.

B. MANUSCRIPT ORGANIZATION AND FORMATTING:

Sincerely,

Reilly Lorenz
Editorial Office Life Science Alliance
Meyerohofstr. 1
69117 Heidelberg, Germany
t +49 6221 8891 414
e contact@life-science-alliance.org
www.life-science-alliance.org

3rd Editorial Decision14 August 2020

RE: Life Science Alliance Manuscript #LSA-2020-00788-TRR

Prof. Volker Gerke
University of Muenster
Institute of Med. Biochemistry
Von-Esmarch-Str. 56
Muenster D-48149
Germany

Dear Dr. Gerke,

Thank you for submitting your Research Article entitled "Plasma membrane phosphatidylinositol (4,5)-biphosphate promotes Weibel-Palade body exocytosis". It is a pleasure to let you know that your manuscript is now accepted for publication in Life Science Alliance. Congratulations on this interesting work.

DISTRIBUTION OF MATERIALS:

Again, congratulations on a very nice paper. I hope you found the review process to be constructive and are pleased with how the manuscript was handled editorially. We look forward to future exciting submissions from your lab.

Sincerely,
Shachi

Shachi Bhatt
Executive Editor

Life Science Alliance
www.life-science-alliance.org